# Quantification of the Tissue Oxygenation Delay Induced by Breath-Holding in Patients with Carotid Atherosclerosis

**DOI:** 10.3390/metabo12111156

**Published:** 2022-11-21

**Authors:** Andrés Quiroga, Sergio Novi, Giovani Martins, Luis Felipe Bortoletto, Wagner Avelar, Ana Terezinha Guillaumon, Li Min Li, Fernando Cendes, Rickson Coelho Mesquita

**Affiliations:** 1“Gleb Wataghin” Institute of Physics, University of Campinas, Campinas 13083-859, SP, Brazil; 2Brazilian Institute of Neuroscience and Neurotechnology, Campinas 13083-970, SP, Brazil; 3Clinical Hospital, University of Campinas, Campinas 13083-888, SP, Brazil; 4Faculty of Medical Sciences, University of Campinas, Campinas 13083-894, SP, Brazil

**Keywords:** carotid artery stenosis, functional near-infrared spectroscopy, tissue oxygenation, cerebral vasoreactivity, breath-holding

## Abstract

Carotid artery stenosis (CAS) is a common vascular disease with long-term consequences for the brain. Although CAS is strongly associated with impaired cerebral hemodynamics and neurodegeneration, the mechanisms underlying hemodynamic impairment in the microvasculature remain unknown. In this work, we employed functional near-infrared spectroscopy (fNIRS) to introduce a methodological approach for quantifying the temporal delay of the evoked hemodynamic response. The method was validated during a vasodilatory task (breath-holding) in 50 CAS patients and 20 controls. Our results suggest that the hemodynamic response to breath-holding can be delayed by up to 6 s in the most severe patients, a significant increase from the median 4 s measured for the control group (*p* = 0.01). In addition, the fraction of brain regions that responded to the task decreased as the CAS severity increased, from a median of 90% in controls to 73% in the most severe CAS group (*p* = 0.04). The presence of collateral circulation increases the response to breath-holding and decreases the average time delays across the brain, although the number of communicating arteries alone cannot predict these fNIRS-based hemodynamic variables (*p* > 0.09). Overall, this work proposes a method to quantitatively assess impaired cerebral hemodynamics in CAS patients.

## 1. Introduction

The carotid arteries are highly prone to developing atherosclerotic plaque at the origin of the internal carotid artery. Carotid artery stenosis (CAS) is a common atherosclerotic condition with a prevalence estimated to be 1.5% worldwide [1] and a known risk factor for stroke [2,3,4,5]. Studies have shown that CAS causes hemodynamic impairments in multiple domains, which leads to neurodegeneration, even in asymptomatic patients [6,7,8,9]. Cerebral vasoreactivity tests are commonly used to assess hemodynamic impairment by exploring the ability of the brain’s vasculature to increase cerebral blood flow (CBF) in response to a vasodilatory stimulus [10,11,12]. In CAS patients, the pathophysiologic mechanism of inducing arteriolar dilation to maintain cerebral perfusion in an environment of reduced large artery flow leads to a permanent condition of low cerebrovascular reserve (CVR) and increased oxygen extraction fraction (OEF) [13,14]. A variety of possible stimuli can be used to induce vasodilation, including breath-holding, inhalation of 5% CO_2_, or administration of carbonic anhydrase inhibitors such as acetazolamide. Literature suggests that all methods are well correlated [12,13]; among those, breath-holding tests are easy to perform, and there are no associated costs.

Techniques that can directly or indirectly measure the cerebrovascular status during a vasoreactivity test can help the prognosis and therapeutics of patients with CAS. The cerebral blood flow velocity in the middle cerebral artery has traditionally been measured with transcranial Doppler ultrasound (TCD) to assess CVR. In fact, impairments following a vasodilatory stimulus in CAS patients have been previously described using TCD [6,7,15,16]. However, information on blood flow in large arteries alone cannot distinguish reduced CBF caused by narrowing the carotid arteries from compensatory physiological reductions in CBF caused by reduced metabolic demands. In addition, since compensatory mechanisms can be local, the measurement of a large vessel will not be specific to regional hemodynamic variations across the brain.

In this context, functional near-infrared spectroscopy (fNIRS) is an alternative technique that can provide complementary information about the hemodynamic status in the brain by measuring blood oxygen saturation noninvasively and in a portable manner [17,18,19,20,21,22,23,24]. Briefly, fNIRS employs near-infrared light (~700–900 nm) to measure oxy- (HbO) and deoxy-hemoglobin (HbR) concentration changes at the local microvasculature. Total hemoglobin concentration (HbT) changes, which can be readily estimated by adding HbO and HbR, are proportional to changes in cerebral blood volume at the measured region of interest. fNIRS has been previously validated and applied to CAS patients during a cerebrovascular reactivity test under different stimuli [25,26,27,28,29,30]. Of particular interest, most previous studies use the breath-holding index (BHI, defined as the change between the measurement at the end of breath-holding and baseline, normalized by the duration of the breath-holding and the baseline measurement) to characterize cerebral vasoreactivity. While this analysis is sensitive to the severity of the stenosis [30] and correlates with TCD both in the affected and nonaffected carotid arteries [26,28], it ignores the temporal dynamics of the induced vasodilation. Furthermore, when fNIRS data is available in more than one location, they are typically collapsed into one single average across different areas of the head. Although cerebral vasoreactivity stimulation is supposed to be a global effect throughout the brain, it may not have the same impact on every brain region.

This work aimed to analyze the temporal dynamics and spatial variability of the fNIRS-based cerebral blood volume response to voluntary breath-holding in CAS patients. We hypothesized that different brain regions fed by the internal carotid artery can respond differently to vasodilatory stimulation and that the dynamics of this response can be related to the severity of stenosis. In particular, we quantified the hemodynamic dynamics by the temporal delay between the start of the breath-holding and the beginning of the hemodynamic response and compared it among different brain locations to understand how cerebrovascular reactivity varies throughout the regions fed by the internal carotid artery.

## 2. Materials and Methods

### 2.1. Patients

We considered 50 patients diagnosed with CAS and 20 controls for this study. All patients were recruited at the Neurovascular Clinic of the Hospital of the University of Campinas (Brazil). Patients with stroke history or dementia, diagnosed with atrial fibrillation or congestive cardiac insufficiency, cranial defects that can affect fNIRS measurements, and pregnant women, were excluded before enrollment, as well as those who were unable to hold their breath for the target period. The control group was formed by some of the patients’ accompanying persons and mostly consisted of relatives (husband/wife, son/daughter). The local Research Ethics Committee of the University of Campinas approved the experimental protocol (CAAE: 31592420.3.0000.5404). All participants provided written informed consent before any measurement was performed.

Medical history and a neurological examination were performed on all patients to exclude a previous and recent stroke or transient ischemic accident record (<1 year). The CAS diagnosis was initially made with carotid ultrasound and then confirmed with CTA of cervical vessels or digital arteriography. Based on the CTA/angiography, each carotid artery was classified according to the level of stenosis: <50% (considered normal), 50–69%, 70–90%, and >90% (assumed as occluded, i.e., when there was no detectable patent lumen at gray-scale ultrasound).

The demographics of all recruited patients in this study are shown in Table 1, separated by groups. For most of our research purposes, patients were grouped as unilateral or bilateral stenosis depending on the number of carotid arteries with stenosis. Patients were further classified for correlation analysis with the disease severity, considering the stenosis level of both hemispheres. Unilateral patients were classified with severity level 1 (one normal hemisphere and another one with 50–69%) and 2 (one normal hemisphere and another with 70–90%). Bilateral patients were classified with severity level 3 (one hemisphere with 50–69% stenosis and another with 70–90%) and 4 (all cases with stenosis levels higher than 70% in both hemispheres). Controls were assigned a severity level of zero.

### 2.2. fNIRS Data Acquisition

The fNIRS probe was designed to cover the main brain regions irrigated by the carotid arteries with 14 sources (each source had two different wavelengths, 760 and 850 nm) and 30 detectors arranged in a head cap (Figure 1a). This geometric configuration allowed 48 different source–detector pairs (channels) with a 3 cm separation between sources and detectors. All fNIRS measurements were performed with a commercial system (NIRScout, NIRx Medical Technologies, Germany) with an acquisition rate of 8.2 Hz.

The experimental protocol was carried out during the patient’s regular visit to the clinic. After a 60 s period of rest, subjects were required to perform up to seven trials of voluntary breath-holding (BH) for a target duration of 10 s each. Each trial was followed by a resting period of 30 s (Figure 1b).

### 2.3. fNIRS Data Processing

For each participant, we first discarded channels with a signal-to-noise ratio (SNR) lower than eight from the raw intensity data since they did not contain relevant information. Light intensity from the remaining channels was converted to optical density and then corrected for motion artifacts with a hybrid algorithm [32]. Next, we calculated hemoglobin concentration changes using the modified Beer–Lambert law with a differential pathlength factor of 6 for both wavelengths. Data were band-pass filtered between 0.01 and 0.5 Hz to remove any long-term trends and high-frequency systemic physiology contributions before analysis. Total hemoglobin concentration (HbT) changes were calculated as the sum of HbO and HbR concentration changes. Since breath-holding is a global vasodilatory task supposed to increase blood volume, we focused our analysis on HbT.

The temporal dynamics of each channel following breath-holding were assessed by block-averaging all trials from a single channel from 5 s before the start of the challenge until 35 s after the breath-holding onset (or, equivalently, 25 s after the challenge was completed). Hemispherical time courses were extracted by averaging the temporal courses of all channels with a significant response in the hemisphere, while the subject response was obtained by averaging all channels with a significant response from both hemispheres. Finally, the time course of the group was obtained by averaging the response across all subjects within the group.

### 2.4. Quantification of the Hemodynamic Delay

To locate the hemodynamic response evoked by breath-holding, we performed a general linear model (GLM) on the HbT signal of each channel independently by convolving a canonical hemodynamic response function (HRF) with a boxcar with the breath-holding duration. We assumed the HRF as a gamma function without an undershoot defined by [33]:(1)$$\mathrm{HRF}\left(t\right)=\frac{t^{\left(a-1\right)}}{\Gamma (a)\cdot \tau_{h}{}^{a}}\cdot e^{-t/\tau_{h}},$$
where $\tau_{h}$ is the response width (traditionally set to 1 s) and a represents the peak time. In this study, we assumed a=6 while $\tau_{h}$ arbitrarily varied from 1 to 5 s. Our analysis purposely changed $\tau_{h}$ to adapt the model to the altered dispersion in the cerebrovascular hemodynamics observed in patients with atherosclerosis in the analysis of the temporal dynamics (see Section 3.1).

The proposed adaptive general linear model (aGLM) still fits the experimental time series with an HRF defined by two fixed parameters (*a* and $\tau_{h}$), similar to the regular GLM. However, in the aGLM, this procedure is repeated independently for different values of $\tau_{h}$; channels that exhibited significant HbT increases compared to the baseline (*p* < 0.05) for any $\tau_{h}$ in the HRF were considered a response to breath-holding. When a single channel presented a vasodilatory response for more than one response width, we chose the $\tau_{h}$ that minimized the error between the HRF model and the experimental data.

Since the response width affects the dynamics of the hemodynamic changes during breath-holding, the proposed aGLM provides insight into the microvascular impairment in CAS patients. The impairment degree can be quantified through the time delay between the breath-holding onset and the start of the cerebrovascular vasodilatory response evoked by the challenge (Figure 2). By varying the response width between 1 and 5 s, we could fit hemodynamic responses with onset delays ranging from 1 s to 7 s post-breath-holding. (Note, we arbitrarily assumed the HRF onset as the time the HRF amplitude was larger than or equal to 0.01.) Therefore, the aGLM approach allows both the identification of the brain regions associated with the vasodilatory task and the estimation of the oxygenation delay following the vasodilatory task in each channel.

The median and interquartile range (IQR) of the estimated delays representing the slower dynamics across all channels that exhibited a response to breath-holding in each hemisphere were used for hemispherical analysis in every participant. For time delay analysis, we assigned the maximum time delay (7 s) to the channels that did not achieve statistical significance in the aGLM model across the entire range of investigated response width. Group averages across all subjects within the group were taken by averaging the temporal delays for each subject channel.

### 2.5. Statistical Analysis

To compare the groups, we extracted the parameters of interest for each subject (or hemisphere). Given the heterogeneity of the disease and the small sample size to account for the high variability in each group, we performed nonparametric statistical tests to assess differences across groups. Specifically, we used a paired two-sided Wilcoxon rank sum test for the spatial analysis of activation to evaluate whether two groups had equal medians. A one-sided Wilcoxon rank sum test assessed whether HRF delays were longer in the stenosis groups than in controls. For the temporal analysis, we extracted the peak (positive or negative) of every time course and its corresponding time for each subject. Then, a one-sided Wilcoxon rank sum test was used to compare the parameters extracted from the time series between the two groups.

## 3. Results

### 3.1. Temporal Dynamics of the Hemodynamic Response to Breath-Holding

Figure 3 Group averaged time-courses of the oxy-hemoglobin (HbO, red), deoxy-hemoglobin (HbR, blue), and total hemoglobin (HbT, green) concentration changes due to the breath-holding task performed by (a) control, (b) unilateral, and (c) bilateral groups. The solid curves represent the median values across all subjects in each group, while the shadows around the curves represent the standard error of the mean. The gray shadow in the plots represents the task durationshows the average hemoglobin concentration time series across all channels around the breath-holding task for all groups. Although the hemodynamic response is quite heterogeneous within groups (reflected by large standard errors represented by the shaded areas), it is possible to observe significant differences in the long-term trends for all contrasts (i.e., HbO, HbR, and HbT) that are potentially related to differences in the cerebrovascular reactivity among the groups.

The control and unilateral groups exhibit the temporal response expected for breath-holding experiments typically reported in the literature with fNIRS [34,35,36,37,38]. The dynamics of the time courses can be easily interpreted since holding the breath increases the partial pressure of CO_2_ (pCO_2_), a cerebral vasodilator. The vasodilation leads to increased cerebral blood volume across the global vasculature, reflected by the rise in HbT compared to the baseline. Since the dilation occurs primarily in the arterial side of the vasculature, the larger amount of blood volume is accompanied by an increase in blood oxygenation, leading to higher levels of HbO concentration than the baseline. The increase in HbR measured in fNIRS during breath-holding is harder to interpret since it results from the interplay between vasodilation in cerebral and extracerebral tissues and the metabolism required by the smooth muscle cells to dilate the extracerebral vasculature.

The control group showed a slow but robust increase in HbR, reaching a peak of 20.5 s after the challenge onset (Figure 3a). The higher HbR compared to baseline is accompanied by similar increases in HbO and HbT that reach their maximum at 13.2 s and 14.3 s after the challenge onset, respectively. These two chromophores peak slightly longer in the unilateral group (Figure 3b), at 15.0 s and 16.0 s, respectively, although the difference is not statistically significant considering all subjects (*p* = 0.36 and *p* = 0.59 for HbO and HbT, respectively). At the same time, the increase of HbR in the unilateral group reaches its maximum at 18.9 s, which is faster than the control group (*p* = 0.05). Although not significant, these differences are consistent with the hypothesis that ICA stenosis restricts blood availability at the microvasculature. The faster HbR dynamics in unilateral stenosis could be associated with the difficulty of balancing metabolic demand and vasodilation.

On the other hand, the average time course of the bilateral group shows a slight and continuous decrease in blood oxygenation (HbO) in parallel with an increase in HbR, resulting in a slight change in blood volume (Figure 3c). These dynamics were seen for most patients within the bilateral group, although few showed an HRF with a similar time course from the unilateral and control groups. The behavior exhibited in Figure 3c would be observed in the face of increased OEF when the oxygen available at the local microvasculature is used, but there is not enough supply of blood to match the demand required by the vasodilatory challenge. Unlike the previous two groups, both HbO and HbR are significantly different from the baseline at the end of the time window analyzed (25 s after the end of the challenge) in the bilateral group. By interpolating the time series, we projected HbR to take 52 s to return to baseline (not shown in Figure 3c). Similarly, the decrease in HbO in the bilateral group would last approximately 80 s from the beginning of breath-holding. Compared to HbR, the longer time of HbO to return to baseline can indicate the limited vasodilatory capability expected in bilateral patients.

When comparing the average temporal response measured in each hemisphere across the two patient groups, we did not find any significant differences between the two hemispheres due to the vasoreactivity task (Figure 4). The hemispherical similarity was already expected for the control group, and it works as a controlled condition in this case.

Overall, the differences in the temporal dynamics induced by breath-holding encourage a personalized approach to quantify the vasodilatory responses at different time delays since a standard GLM approach would yield no significant responses due to breath-holding for the bilateral group.

### 3.2. Characterization of the Hemodynamic Response Delays across Groups

Figure 5 shows responses to breath-holding at the individual level for arbitrary participants separated by group. It is clear from this overall sample that the responses are quite heterogeneous within each group. We observed a global response (i.e., a significant increase in HbT across all channels) with a short time delay uniform across the brain regions for most controls. Few subjects in the control group showed a global and uniform but delayed response, while others had local delays in specific areas (mainly in the inferior frontal cortex).

The response in the unilateral group was also predominantly global but nonuniform. The time delays of the vasodilatory response were quite variable across the brain for most unilateral patients, with most responses delaying 3–6 s. This pattern was also observed in the bilateral patients. However, almost half of the patients in this group lacked any vasodilatory response in a significant portion of the brain, one pattern that was rarely observed in the unilateral group.

We quantified the cerebrovascular capacity to vasodilate following breath-holding by counting the fraction of channels with a significant response in the fNIRS probe (β>0 and p≤0.05), independently of the time delay. We observed that the distribution of the active channels per subject for each patient group is bimodal, and the more severe the condition is, the more significant the difference between the two distributions. This difference is reflected in the lower fraction of active channels during breath-holding for the bilateral group (median (IQR) = 78 (33)%) when compared to either the unilateral patients (92 (21)%) or the controls (90 (18)%), which is probably associated with the inability of the bilateral patients to respond to a vasodilatory challenge. Given the data distribution, we have not performed any statistical tests on the number of activated channels between groups.

Concerning the time delays, patients appear to take slightly longer to vasodilate than the controls. The hemodynamic onset following breath holding took a median (IQR) of 4.7 (1.6) s in the patient group, which is approximately 15% longer than the average delay found in controls (4.0 (1.4) s, *p* = 0.04). When split between unilateral and bilateral stenosis, the hemodynamic latency was slightly longer for the latter: 4.4 (1.6) s and 4.9 (1.8) s for the unilateral and bilateral groups, respectively. The difference in the response delay was only significant for the bilateral group compared to the control subjects (*p* = 0.16 and 0.01 for the unilateral and bilateral groups, respectively).

### 3.3. Influence of CAS Severity on the Hemodynamic Response Delays across Groups

The high variability observed in the time delays potentially reflects the broad range of stenosis levels within each patient group. To investigate this problem further, we decided to further split the groups according to the severity of stenosis. The descriptive statistics of all the fNIRS-based hemodynamic parameters measured with our approach during breath-holding are summarized in Table 2.

With this approach, we observed that all the hemodynamic response features measured with fNIRS depend on the stenosis severity. The number of brain regions that responded to the vasodilatory task decreased as the stenosis levels increased (Figure 6a), consistent with lower cerebrovascular reserve observed for these patients. However, the difference was not statistically significant when comparing any groups, probably due to the high heterogeneous distribution, which increased with the CAS severity.

The increase in heterogeneity for the more severe patients can be partially explained by collateral circulation. It is evident from Table 3 that more functional collaterals develop as the disease progresses in an attempt to ameliorate the hemodynamic impairment caused by the stenosis. However, these collateral pathways do not happen similarly to all patients in each group, which leads to heterogeneity for the more severe patient groups.

Similarly, we found a strong dependency of the time delay between the breath-hold onset and the hemodynamic response averaged across all different brain regions with fNIRS (Figure 6b). Although the time delay of the mildest CAS patients did not significantly differ from the control group (*p* = 0.69), all other patients showed longer time delays than controls. Unilateral patients with 70–90% stenosis took a median (IQR) of 4.8 (1.5) s to respond to breath-holding, which is significantly longer than both the control group (*p* = 0.03) and mild unilateral patients (*p* = 0.01). The hemodynamic response was progressively slower for the bilateral patients, taking 4.9 (1.7) s and 5.0 (1.7) s for CAS severity 3 and 4, respectively. In extreme cases, we observed average time delays longer than 6 s for approximately 26% of the bilateral CAS patients, while only one unilateral CAS patient (4%) showed such a long delay.

We also investigated the dispersion (quantified as the standard deviation) of the time delays across the channels that exhibited a vasodilatory response since it could provide information about the heterogeneity of the vascular response induced by vasoreactivity (Figure 6c). Overall, patients showed a slightly higher standard deviation of time delays than controls (*p* = 0.08). However, the heterogeneity of the response did not increase as a function of CAS severity when we separated the patients into different groups. The three most severe CAS groups showed similar median (IQR) dispersion of 1.3 (0.6), 1.1 (0.9), and 1.3 (0.6) for CAS severity 2, 3, and 4, respectively, while the dispersion of the control group and the mildest CAS severity was 1.1 (0.7) and 0.9 (1.1), respectively.

Since the brain hemispheres and their vascularization are not symmetric, we attempted to quantify any potential hemispheric differences during vasodilation by computing the laterality index for both the fraction of responded channels and the time delays (i.e., LI = (H1–H2)/(H1 + H2), where H1 and H2 represent the two cerebral hemispheres). In our analysis, H2 was always considered as the hemisphere ipsilateral to the ICA with stenosis/occlusion (unilateral group) or the ICA with the highest level of occlusion (bilateral group); therefore, negative values of LI suggest longer delays in the hemisphere ipsilateral of ICA (unilateral group) or the highest level of ICA (bilateral group). Overall, we found no significant differences between the two hemispheres (Figure 7). Even when we found a delayed vasodilatory response to breath-holding in a given brain region, the homotopic area showed a similar delay. As a result, the laterality index was homogeneously distributed around zero in all cases, with few outliers in the patient groups.

### 3.4. Modeling of the Hemodynamic Response Delays as a Function of the CAS Severity

Last, we employed linear regression analysis to independently model the fraction of activated channels and the average time delays as a function of both stenosis severity and/or the presence of collateral pathways, quantified by the number of opened communicating arteries across anterior and posterior circulation. In all cases, the hemodynamic parameter was modeled as a linear function of either the CAS severity, the number of communicating arteries providing collateral circulation, and the interaction of these two parameters (the relation was quantitatively assessed with a coefficient, *b*, relating each dependent variable with an independent variable).

Our results showed a moderate correlation between the stenosis severity with both fNIRS parameters: the higher the severity, the lower the number of activated channels (*r =* −0.26, *p* = 0.04) and the higher the temporal delay between the vasodilatory challenge and the hemodynamic response (*r =* 0.33, *p* = 0.01). Furthermore, stenosis severity can predict time delay (*b* = 0.34 (0.15), *p* = 0.025) but not with the fraction of activated channels (*b* = −2.1 (1.5), *p* = 0.11). Of note, the time delay estimated for CAS severity of zero (i.e., the model’s intercept) was 3.2 (0.7), within the time delay interval estimated for both the control and the low CAS severity groups.

On the other hand, collateral circulation was only weakly associated with both the average time delay (*b* = −0.14 (0.21), *p* = 0.52) and the fraction of activated channels (*b* = 3.3 (2.1), *p* = 0.09). When all patients were considered, the number of communicating arteries was poorly correlated with both the average time delay (*r =* −0.09, *p* = 0.28) and the number of activated channels (*r = 0.23*, *p* = 0.08). The lack of an overall pattern is probably related to the variability in the development of collateral pathways, as previously appointed.

## 4. Discussion

In this work, we describe a novel fNIRS-based approach for assessing the vasodilatory response of the local cerebral microvasculature in patients diagnosed with carotid atherosclerosis (CAS). Currently, fNIRS analysis typically uses the general linear model (GLM) to find the evoked hemodynamic response function (HRF) associated with a given task. Standard GLM, however, assumes a priori HRF with fixed dynamics (commonly inferred from healthy individuals) and ignores the possibility that the response may be delayed under certain conditions (e.g., cerebrovascular diseases). To address this issue, we demonstrate a method of accounting for temporal mismatches in the evoked response. This methodology can be valuable for properly translating fNIRS to clinical research by providing more accurate information about the evoked hemodynamics in brain-injured populations.

By varying the dynamics of the HRF used in the standard GLM analysis, we were able to find the optimum response width that best matched the experimental data. This adaptive GLM (aGLM) strategy allowed us to quantify the slower hemodynamics observed in CAS patients rather than assuming they did not respond to the vasodilatory task. The use of aGLMs as a methodological technique to optimize evoked brain function has been previously suggested in the literature [39] but has not been translated into meaningful clinical information. As shown here, the aGLM can provide a parameter for quantifying delayed vasodilatory responses across brain regions.

The proposed methodology was validated with an induced vasoreactivity protocol. In our opinion, quantifying the time delays appears to be a valuable tool to better understand the hemodynamic impairments in CAS patients. For the purpose of this experiment, we asked participants to hold their breath for 10 s to determine cerebrovascular reactivity. Compared to clinical protocols, our task duration is shorter than the target 30 s commonly used with TCD. However, our practical experience was that very few elderly participants could actually hold their breath for that long. In our protocol, we confirmed that all participants recruited in this study were able to hold their breath for at least 10 s, which gave us more confidence that any conflicting result would not be due to variability in the experimental protocol. Furthermore, we hypothesized that hemodynamic changes in local microvasculature assessed by fNIRS could be detected with a shorter duration since the relative response to breath-holding would be greater than that in larger vessels measured with TCD. The average time courses support our hypothesis, and we were able to successfully see a significant response correlated with breath-holding at the group (Figure 3 and Figure 4) and the individual (Figure 5) levels.

Like previous studies that performed breath-holding with fNIRS, our experimental setup only used long source–detector separations (3 cm) [26,34,40,41,42]. Therefore, the optical signal analyzed in this work contains both cerebral and extracerebral contributions, which makes the physiological interpretation of HbO and HbR changes challenging since the interplay of oxygen supply and demand will have different dynamics at different depths underneath the scalp. The inability to unentangle cerebral and extracerebral contributions with our fNIRS setup was one of the main reasons we focused on HbT changes. The vasodilation induced by the increase in pCO_2_ during breath-holding should affect all arterial vessels (i.e., cerebral and extracerebral) at different levels but in the same direction to increase blood volume, which would be observed as an increase in HbT everywhere in the brain and outside. On the other hand, there would be no reason to expect that the vasodilatory mechanisms would coincide at the extracerebral and the cerebral microvasculature. Feedback and feedforward signaling of the neurovascular unit suggest that cerebral vasoreactivity will most likely yield different dynamics at a given instant, significantly affecting the temporal dynamics measured by fNIRS. On top of that, it is expected that a disease of vascular origins, such as CAS, will affect how blood responds to a vasodilatory task and introduce temporal shifts in blood flow and oxygenation throughout the cerebrovasculature. To the best of our knowledge, it is surprising that these effects have never been considered in previous fNIRS studies with CAS patients.

The results presented in this work reinforce that the vasodilatory response of CAS patients is constrained in several ways, ranging from a delayed to a complete lack of hemodynamic response. Moreover, we found a significant correlation between vasodilating capability and CAS severity, with CAS severity associated with reduced vasodilating capability (Table 2). We quantified this observation more accurately by counting the number of fNIRS channels that showed a vasodilatory response following breath-holding at different time delays (Figure 6a). The lower number of channels with a vasodilatory response in the more severe forms of CAS is consistent with the hypothesis that permanent vasodilation acts as a compensatory mechanism for the lower blood supply provided by the carotid arteries [43,44,45]. It is worth noting that, although breath-holding is a global disturbance expected to affect the whole vasculature, not all fNIRS channels yielded a significant change correlated with the task, even in the control group. The lack of local response may be related to the amount of disturbance caused by vasodilation, which can vary in different brain locations. From the technological perspective, the SNR of channels located in the various areas will also fluctuate. The physiological and technical aspects of the measurement will lead to regions in which our fNIRS probe will not have enough sensitivity to identify changes associated with breath-holding robustly.

The hemodynamics was significantly altered in the remaining brain regions that showed an evoked response to breath-holding. We found that more severe stenosis leads to longer average delays in the cerebral hemodynamic response across the brain during the vasodilatory task (Figure 6b). However, the delay does not appear to affect all regions homogeneously, as the dispersion of time delays also increases with CAS severity (Figure 6c). Notably, the average time delays extracted by our methodology are associated with stenosis severity (Section 3.4), suggesting that this parameter holds potential for clinical assessment of CAS.

Across all data analyses, we observed significant heterogeneity within each group, which ended up limiting the power to assess the statistical significance of our sample. There are several sources for this heterogeneity. First, introducing a new free parameter in the analysis (time delay) will naturally allow more intrinsic variability in the results and contribute to our reported overall heterogeneous patterns. On top of this inherent variability, different reasons will increase the variability within the group. The participants in the control group were randomly selected based on availability to match the age and sex of the patient group. Since we did not perform any diagnostic exam for the presence of atherosclerotic plaques at the ICA for the control group, some of the controls may have asymptomatic impaired hemodynamics, which could explain the few cases with higher time delays found in this group.

In the patients’ groups, the presence of the atherosclerotic plaques will eventually force the cerebrovasculature to reorganize itself and develop new functional mechanisms that will compensate for the lower level of blood going into the brain and therefore contribute to cerebral autoregulation (as poor cerebral collateral circulation is a common risk factor for ischemic stroke [46]). This phenomenon is evident after quantifying the number of communicating arteries providing collateral blood flow, which is more heterogeneous for the most severe CAS groups (Table 3). However, we found weak association between collateral circulation and the average delay or the fraction of responding brain regions across different groups. The poor correlation can be explained by the fact that the compensatory mechanisms are expected to differ across patients even within the same group, leading to a unique cerebrovascular reorganization with variable effects in the local microvasculature. Therefore, the variability in collateral flow will naturally lead to heterogeneity in brain hemodynamics parameters measured with fNIRS and limit any group analysis. In fact, group averaging may not be the best strategy for understanding the effects of CAS at the microvascular level and translating findings into clinical care. The methodology presented in this work may open new directions toward the individualized characterization of cerebrovascular deficits due to CAS, addressing a complex public health problem that currently lacks a practical solution.

Since each ICA feeds a different hemisphere, one may expect hemispherical asymmetries in the vasoreactivity response to breath-holding. The disparity between the two hemispheres would be even more expected in the unilateral group since one of the hemispheres provides normal flow to the brain. However, we could not find any observation that supports this hypothesis. Instead, we observed similar characteristics in both hemispheres when comparing one given measured region with its homotopic area at the contralateral side. Again, the collateral circulation at the circle of Willis might explain the similarity across both hemispheres. It has been shown that the anterior communicating arteries provide collateral channels in patients diagnosed with unilateral ICA [47]. In addition, the carotid blood flow contralateral to the stenosis hemisphere increases to compensate for the decreased perfusion on the ipsilateral side [48]. However, the posterior circulation appears to be demanded entirely only after severe stenosis occurs [47,48], which is the case in patients with bilateral carotid stenosis. All these factors will contribute to the reorganization of the cerebral vasculature, which will act collectively as a highly connected system to minimize impaired brain function at the microvasculature.

Even though some general limitations have already been highlighted throughout the discussion, there are still some to consider. First, as our primary objective was to demonstrate a new methodological approach, we validated our approach using all CAS patient results without attempting to influence clinical interpretations. The development of quantitative guidelines about temporal delays that can be translated to patients requires a clinical research study with a large cohort and adequate sample size estimations in the future. As a result of the larger cohort, we would also be able to analyze the role of collateral circulation within each group more thoroughly with an individualized approach. Regarding methodological significance, dividing cerebral contributions from extracerebral contributions would be valuable as this could provide an interpretation of the temporal delays based on how the brain works under restricted flow conditions. In the future, we plan to address these critical issues to provide clinicians with a reliable tool to better understand the mechanisms underlying carotid artery stenosis.

## 5. Conclusions

This work introduced and validated a methodological approach to quantitatively assessing hemodynamic impairment at the microvasculature during a vasodilatory task. The proposed method provides novel insights into the temporal hemodynamic mismatch following breath-holding in CAS patients. Specifically, the time delay clarifies the vasodilatory processes induced by breath-holding (cerebral and extracerebral changes are yet to be studied). We observed that CAS patients take longer than controls to vasodilate and have a more diverse response across different brain areas. Quantification of activated channels in bilateral patients is a measure of vasodilatory capacity and may be used to quantify CAS severity. We observed no hemispherical variations in local microvascular reactivity in any groups. In the future, the use of fNIRS montages with short channels in a larger cohort focused more on a clinical trial than on a methodological development will allow us to separate the contributions of the extracerebral from the cerebral vasculature and further examine the interplay of the temporal blood oxygenation dynamics at different locations.

## Figures and Tables

**Figure 1 metabolites-12-01156-f001:**
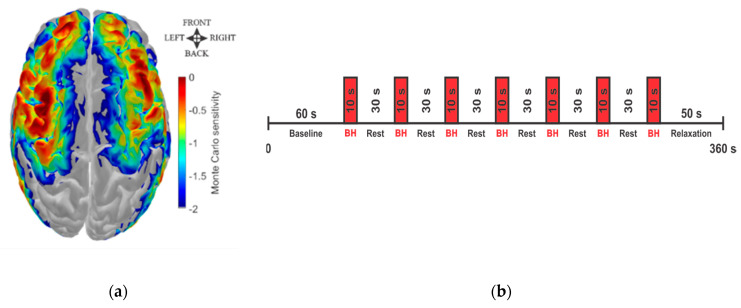
fNIRS Data acquisition. (**a**) Brain regions monitored with fNIRS, covering the main brain areas irrigated by the carotid artery. The color bar represents a map of photon sensitivity based on Monte Carlo simulations performed with AtlasViewer [31]; the red color represents regions with high photon sensitivity, while blue represents low–photon sensitivity. (**b**) Experimental protocol performed by all participants, consisting of up to seven breath-holding trials of 10 s each, followed by 30 s of rest for a total of 6 min of acquisition.

**Figure 2 metabolites-12-01156-f002:**
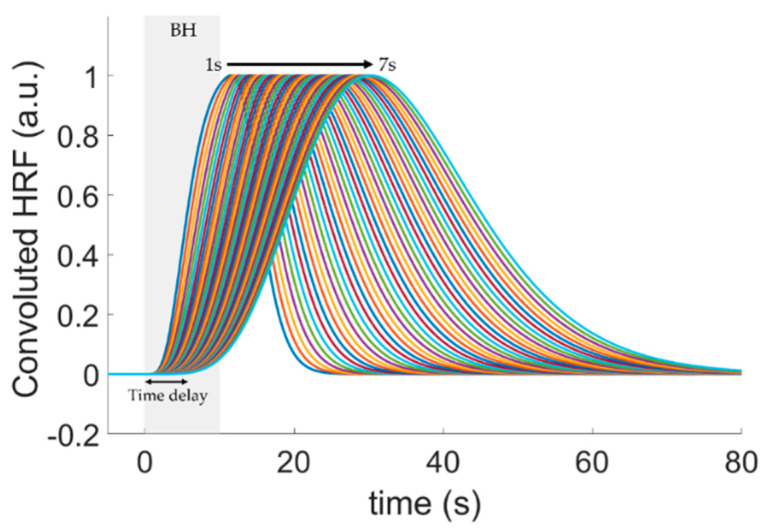
Hemodynamic response function (HRF) models used in the proposed adaptive GLM (aGLM) analysis. By varying the dispersion time of the HRF basis function ($\tau_{h}$ in Equation (1)), we could model differences in the transit time of the cerebrovascular response to breath−holding in patients diagnosed with carotid artery stenosis (CAS). The HRFs with different delays (solid curves) were obtained by convolving the HRF basis function with a boxcar representing the breath-holding challenge (BH). The resulting HRFs were characterized by their delayed onset, ranging from 1 s to 7 s.

**Figure 3 metabolites-12-01156-f003:**
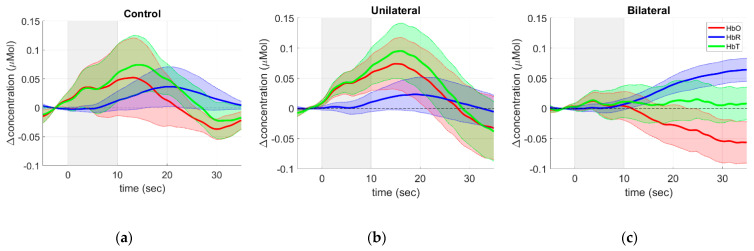
Group averaged time-courses of the oxy−hemoglobin (HbO, red), deoxy−hemoglobin (HbR, blue), and total hemoglobin (HbT, green) concentration changes due to the breath-holding task performed by (**a**) control, (**b**) unilateral, and (**c**) bilateral groups. The solid curves represent the median values across all subjects in each group, while the shadows around the curves represent the standard error of the mean. The gray shadow in the plots represents the task duration.

**Figure 4 metabolites-12-01156-f004:**
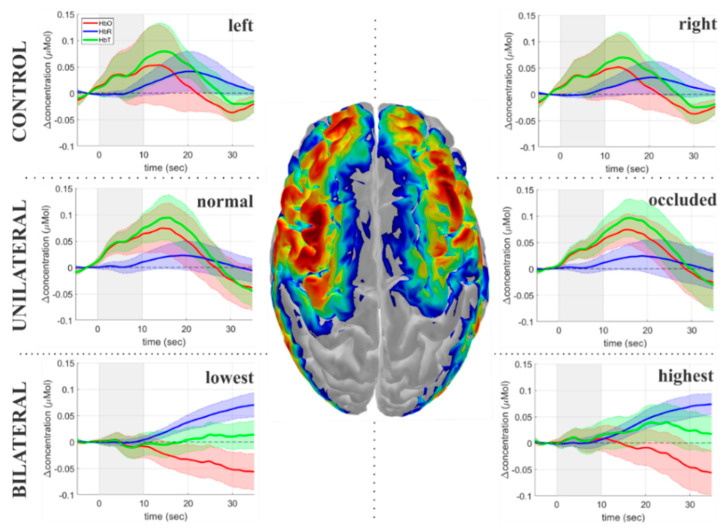
Group averaged time courses for each hemisphere due to the breath−holding task. The plots show the changes in oxy−hemoglobin (HbO, red), deoxy−hemoglobin (HbR, blue), and total hemoglobin (HbT, green) induced by breath−holding for each group. The left column represents the left hemisphere (controls) or the hemisphere ipsilateral to either the nonoccluded ICA (unilateral patients) or the ICA with the lowest level of occlusion (bilateral patients). The right column shows the data from the right hemisphere (controls) or the hemisphere ipsilateral to either the stenotic/occluded ICA (unilateral patients) or the ICA with the highest level of occlusion (bilateral patients). The solid curves represent the median values across subjects, while the color shadows represent the standard error of the mean. The gray shadow in the plots represents the task duration.

**Figure 5 metabolites-12-01156-f005:**
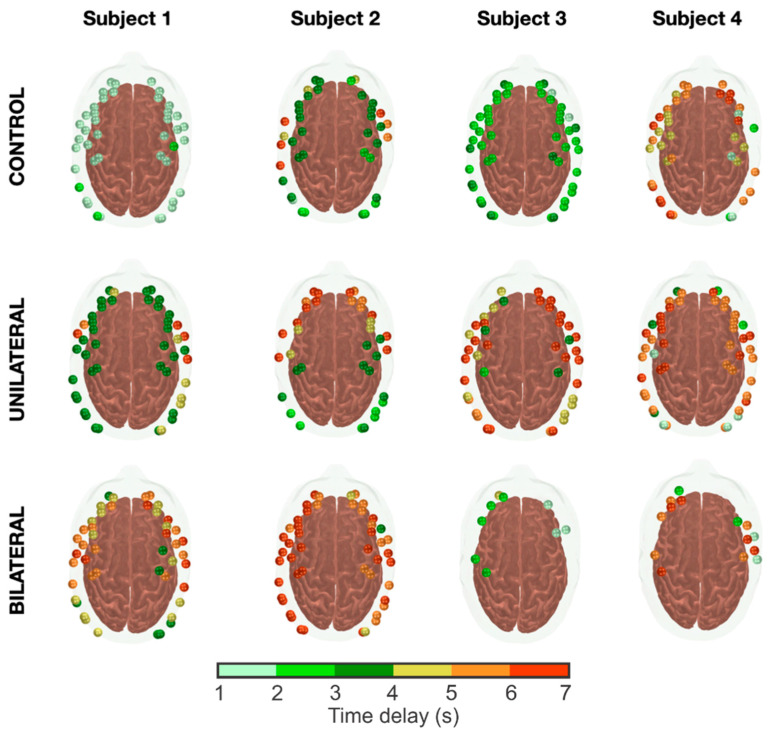
Individual responses to the breath-holding challenge from four arbitrary subjects within each group. Each sphere on the head represents an fNIRS channel that showed a significant vasodilatory response (as measured by a significant increase in HbT) following apnea. The color bar represents the time delays of the modeled HRF that maximizes the fit of the measured time courses.

**Figure 6 metabolites-12-01156-f006:**
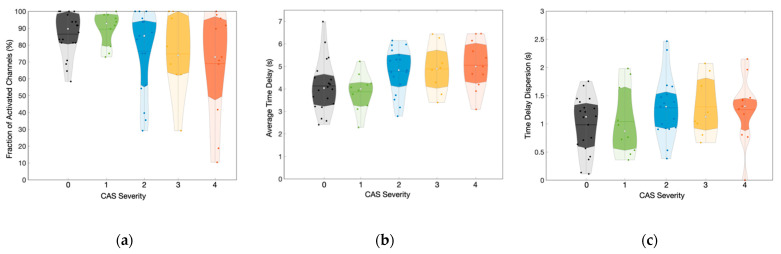
Distribution of the (**a**) fraction of the activated brain channels, (**b**) average time delay across the different brain regions, and (**c**) dispersion of the time delay across the different brain regions as a function of carotid artery stenosis (CAS) severity. Gray: Control, green: unilateral level 1, blue: unilateral level 2, yellow: bilateral level 3 and, orange: bilateral level 4.

**Figure 7 metabolites-12-01156-f007:**
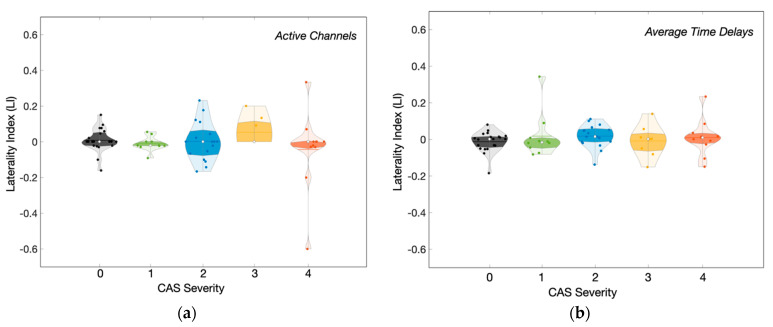
Distribution of the laterality index (LI) as a function of the carotid artery stenosis (CAS) calculated for (**a**) the fraction of active channels that responded to the breath−holding challenge and the (**b**) time delays of the vasodilatory hemodynamic response to breath−holding. Gray: Control, green: unilateral level 1, blue: unilateral level 2, yellow: bilateral level 3 and, orange: bilateral level 4.

**Table 1 metabolites-12-01156-t001:** Demographic and clinical characteristics of the study population. SD: standard deviation, ICA: Internal carotid artery, CCA: Common carotid artery. Data are given as the number of subjects and percentage (i.e., *n* (%)) of the total included subjects within each group, except when indicated in the first column.

Parameter	Patients		Controls
	Unilateral	Bilateral	
Number of subjects	31	19	20
Mean age (years, SD)	67 (8)	68 (7)	63 (8)
Female	7 (22.5%)	5 (26.3%)	6 (30%)
Location of stenosis			
left ICA	9 (29%)	18 (94.7%)	-
right ICA	19 (61.2%)	18 (94.7%)	-
left CCA	2 (6.5%)	1 (5.3%)	-
right CCA	1 (3.2%)	1 (5.3%)	-
Degree of stenosis			
0–49%	1(3.2%)	0(0%)	-
50–69%	9 (29%)	0(0%)	-
70–90%	20 (64.5%)	13 (68.4%)	-
Occluded	1 (3.2%)	6 (31.6%)	-
Presence of collateral circulation			
Anterior communicating arteries	6 (40%)	11 (69%)	-
Right posterior comm. arteries	4 (27%)	6 (38%)	-
Left posterior comm. arteries	3 (20%)	4 (25%)	-
Asymptomatic	9 (29%)	4 (21%)	-
Symptomatic			
Transient ischemic attack	2 (6.5%)	3 (15.7%)	-
Ischemic stroke	20 (64.5%)	12 (63.1%)	-
Other conditions			
Hypertension	27 (87.1%)	16 (84.2%)	10 (50%)
Diabetes	21 (67.7%)	7 (36.8%)	7 (35%)
Smoking	18 (58%)	10 (52.6%)	5 (25%)
Dyslipidemia	19 (61.2%)	13 (68.4%)	4 (20%)
Heart failure	3 (9.6%)	2 (10%)	0 (0%)
Coronary artery disease	2 (6.5%)	4 (21%)	0 (0%)
Chronic kidney insufficiency	2 (6.5%)	3 (15%)	2 (10%)
Etilism	11 (35.5%)	6 (31.5%)	0 (0%)
Obesity	0 (0%)	1 (5.2%)	0 (0%)

**Table 2 metabolites-12-01156-t002:** Features of the hemodynamic response due to the breath-holding challenge as a function of stenosis severity. In all cases, the hemodynamic parameters calculated from fNIRS are shown as the median (25% percentile; 75% percentile) across all subjects within each group.

Group	Stenosis Severity	Number of Subjects	Hemodynamic Features		
			Fraction Activated Channels	Time Delay	Laterality Index (LI)
Control	0	20	90 (81; 99) %	4.0 (3.3; 4.6) s	0.00 (−0.04; 0.02)
Unilateral	1	10	93 (79; 98) %	4.0 (3.3; 4.3) s	−0.02 (−0.05; 0.01)
	2	17	85 (56; 94) %	4.8 (4.1; 5.6) s	0.02 (−0.01; 0.06)
Bilateral	3	9	74 (63; 98) %	4.9 (4.0; 5.7) s	0.00 (−0.07; 0.03)
	4	11	73 (48; 95) %	5.0 (4.3; 6.0) s	0.01 (−0.02; 0.03)

**Table 3 metabolites-12-01156-t003:** Presence of collateral circulation for each stenosis severity. For the patients that had imaging scans, the table shows the number of patients (proportion) with collateral circulation, separated by the total number of communicating arteries with collateral circulation.

Group	Stenosis Severity	Number of Subjects		Total Collateral Circulation		
			0	1	2	3
Unilateral	1	7	2 (29%)	4 (57%)	1 (14%)	0 (0%)
	2	14	5 (36%)	6 (43%)	3 (21%)	0 (0%)
Bilateral	3	8	0 (0%)	5 (63%)	2 (25%)	1 (13%)
	4	9	3 (33%)	4 (44%)	0 (0%)	2 (22%)

## Data Availability

The data presented in this study are available on request from the corresponding author. The data are not publicly available due to the participant consent obtained as part of the recruitment process.

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
