# Peer review of "Quantification of the Tissue Oxygenation Delay Induced by Breath-Holding in Patients with Carotid Atherosclerosis"

_metabolites, 2022, doi:10.3390/metabo12111156_

Round 1

Reviewer 1 Report

The functional near-infrared spectroscopy of the brain can offer infotmation on the cerebral autoregulation mechanisms in the presence of significant  carotid artery stenosis (CAS). The impaired oxygenation, additionally provoked by breath holding test, can be displayed with this method. The test result corresponds to the quality of the regional blood supply within CAS area. The Authors included healthy controls, patients with one-sided CAS and bilateral CAS. 

The Authors have observed huge diversity in vasodilation response across different brain regions as well as in the individual subjects. In addition, the delay in vasodilatatory response was up to 7-fold longer in CAS patients compared to healthy controls. 

My overall impression is that this study is well written. However, my major concern is that it lacks information on the presence or absence of the collateral pathways, e.g. through the anterior communicating and posterior communicating arteries (on TCCD perhaps).  Well developed and functional collaterals contribute to the cerebral  autoregulation, hence decreasing its impairment. On the contrary, poor cerebral collateral circulation is a frequent risk factor associated with ischemic stroke (Badacz R et al. Postepy w Kardiologii Interwencyjnej 2015, 11(4), 312-317.

It would be great if Authors could add information on the collateral flow and compare it with the results of the breath-holding test and vasodilatation, and then address their findings in discussion.

It could be that lack of differences between patients with one-sided versus bilateral CAS is attributed to the presence of functional collateral pathways. 

Minor remarks. Abstract, please give some information on study groups and controls, as well as some numbers in results. now it is not informative at all.

line 164- there is some error in reference

Please consider adding the mentioned above reference to discussion in the context of the ischemic stroke risk and lack of cerebral collateral pathways 

Reviewer 2 Report

The following is the summary of the present study:

Carotid artery stenosis (CAS) is a common vascular disease with long-term consequences for the brain. Evidence suggests that CAS is associated with impaired cerebral hemodynamics and neurodegeneration, even in asymptomatic patients. However, the mechanisms underlying hemodynamic impairment in the microvasculature remain largely unknown. In this work, we employ functional near-infrared spectroscopy (fNIRS) to introduce a novel methodological approach for quantifying the temporal delay of the evoked hemodynamic response. The method was validated during a vasodilatory task (breath-holding) in CAS patients and controls. Our results suggest that the hemodynamic response to breath-holding is highly heterogeneous across different brain regions in CAS patients; compared to controls, the induced vasodilatory response can be delayed up to 7-fold in CAS patients. More importantly, we found that the delayed response is associated with the CAS severity. Overall, this work shows a practical and robust method to assess impaired cerebral hemodynamics in CAS patients quantitatively.

The article is well written. I have some comments:

First, in line 34, please provide a reference for the cerebral vasoreactivity test.

Second, please concentrate the introduction to four paragraphs. The current number of paragraphs makes the whole introduction too piecemeal.

Third, how did the authors determine the number of patients versus the number of controls? Did the authors perform sample size estimation before the study?

Fourth, the number of IRB should be provided.

Fifth, in Table 1, the full form for SD should be added in the legend.

Reviewer 3 Report

Quiroga et al have submitted a manuscript on the method of measuring oxygenation delay caused by breath holding in the cerebral tissue of patients with unilateral or bilateral carotid atherosclerosis. The manuscript is well written, with a thorough methodology and adequate discussions.

Some minor improvements might be brought to the paper to increase its value and interest for the readers:

- how was the size of the control lot chosen and why did you not enlist the same number, or higher, than the study lot?

- please specify whether the classification of the degree of stenosis was done via US or CTA/angio

- do you have any prior publications or literature data to support your proposed aGLM?

- please clarify the Reference source issue

- please add reference for statement at lines 356-357

- a section addressing the study limitations should be introduced, as such, in the discussions chapter

- due to the high amount of data skewing, normalizations, approximations and arbitrary assignments performed in the methodology in order to obtain the data sets it might seem forced to call the method robust, albeit a method validation of sorts was achieved in the study. I would propose the authors to rephrase the first sentence of the conclusions.

Respectfully submitted,

Round 2

Reviewer 1 Report

The Authors answered all comments in satisfactory way. I would like to congratulate on the study